# In Vitro, In Silico and Network Pharmacology Mechanistic Approach to Investigate the *α*-Glucosidase Inhibitors Identified by Q-ToF-LCMS from *Phaleria macrocarpa* Fruit Subcritical CO_2_ Extract

**DOI:** 10.3390/metabo12121267

**Published:** 2022-12-15

**Authors:** Md. Abdur Rashid Mia, Qamar Uddin Ahmed, Sahena Ferdosh, Abul Bashar Mohammed Helaluddin, Md. Shihabul Awal, Murni Nazira Sarian, Md. Zaidul Islam Sarker, Zainul Amiruddin Zakaria

**Affiliations:** 1Department of Pharmaceutical Technology, Faculty of Pharmacy, International Islamic University Malaysia (IIUM), Kuantan 25200, Pahang, Malaysia; 2Drug Discovery and Synthetic Chemistry Research Group, Department of Pharmaceutical Chemistry, Faculty of Pharmacy, International Islamic University Malaysia (IIUM), Kuantan 25200, Pahang, Malaysia; 3Western Pacific Tropical Research Center, College of Natural and Applied Sciences, University of Guam, Mangilao, GU 96923, USA; 4Department of Plant Science, Faculty of Science, Kuantan Campus, International Islamic University Malaysia (IIUM), Kuantan 25200, Pahang, Malaysia; 5Department of Food Science & Nutrition, Faculty of Engineering, Hajee Mohammad Danesh Science & Technology University, Dinajpur 5200, Bangladesh; 6Institute of Systems Biology (INBIOSIS), Universiti Kebangsaan Malaysia, Bangi 43600, Selangor, Malaysia; 7Cooperative Research, Extension and Education Services (CREES), Northern Marianas College, Saipan, MP 96950, USA; 8Borneo Research on Algesia, Inflammation and Neurodegeneration (BRAIN) Group, Department of Biomedical Sciences, Faculty of Medicines and Health Sciences, University Malaysia Sabah, Jalan UMS, Kota Kinabalu 88400, Sabah, Malaysia

**Keywords:** *Phaleria macrocarpa* fruit, subcritical CO_2_ extract, bioactive compounds, *α*-glucosidase, molecular dynamic simulations, network pharmacology

## Abstract

The fruit of *Phaleria macrocarpa* have been traditionally used as an antidiabetic remedy in Malaysia and neighbouring countries. Despite its potential for diabetes treatment, no scientific study has ever been conducted to predict the inhibitor interaction of the protein *α*-glucosidase identified in an extract prepared with a non-conventional extraction technique. Hence, the major aim of this research was to evaluate the in vitro antioxidant, the *α*-glucosidase inhibitors, and the molecular dynamic simulations of the *α*-glucosidase inhibitors identified by Quadrupole Time-of-Flight Liquid Chromatography Mass Spectrometry (Q-ToF-LCMS) analysis. Initially, dry fruit were processed using non-conventional and conventional extraction methods to obtain subcritical carbon dioxide extracts (SCE-1 and SCE-2) and heating under reflux extract (HRE), respectively. Subsequently, all extracts were evaluated for their in vitro antioxidative and *α*-glucosidase inhibitory potentials. Subsequently, the most bioactive extract (SCE-2) was subjected to Q-ToF-LCMS analysis to confirm the presence of *α*-glucosidase inhibitors, which were then analysed through molecular dynamic simulations and network pharmacology approaches to confirm their possible mechanism of action. The highest inhibitory effects of the 2,2-diphenyl-1-picrylhydrazyl (DPPH) radical and *α*-glucosidase on SCE-2 was found as 75.36 ± 0.82% and 81.79 ± 0.82%, respectively, compared to the SCE-1 and HRE samples. The Q-ToF-LCMS analysis tentatively identified 14 potent *α*-glucosidase inhibitors. Finally, five identified compounds, viz., lupenone, swertianolin, *m*-coumaric acid, pantothenic acid, and 8-*C*-glucopyranosyleriodictylol displayed significant stability, compactness, stronger protein-ligand interaction up to 100 ns further confirming their potential as *α*-glucosidase inhibitors. Consequently, it was concluded that the SCE-2 possesses a strong *α*-glucosidase inhibitory effect due to the presence of these compounds. The findings of this study might prove useful to develop these compounds as alternative safe *α*-glucosidase inhibitors to manage diabetes more effectively.

## 1. Introduction

Diabetes mellitus is one of the most common metabolic diseases that continues to be a serious public health concern across the globe and is mostly connected with chronic and abnormal carbohydrate, protein, and lipid metabolisms. Among diabetic patients, more than 90% of patients are affected with type-2 diabetes mellitus (T2DM) which is characterized mainly by insufficient insulin secretion, hyperglycaemia, and insulin resistance. The activity of several enzymatic pathways, including the intestinal *α*-glucosidase enzyme may cause hyperglycaemia [1]. The *α*-glucosidase enzyme aids in the digestion of complex carbohydrates by cleaving oligosaccharides into monosaccharides with the hydrolyses 1,4-*α* glycosidic linkages. It oversees catalysing the last stage in the carbohydrate digestion process, which ultimately leads to postprandial hyperglycaemia in diabetic patients [2]. The inhibitors of the *α*-glucosidase enzyme will compete with the oligosaccharides for the binding site, making them typical competitive inhibitors. The *α*-glucosidase inhibitors will be an excellent medication for postprandial hyperglycaemia treatment [3]. The *α*-glucosidase inhibitors are oral antidiabetic medications that work by inhibiting the digestion of carbohydrates. When this enzyme is inhibited, the rate of carbohydrate digestion slows down, resulting in less glucose absorption as the carbohydrates are not broken down into smaller molecules [4]. There are many different types of synthetic *α*-glucosidase inhibitors currently available in the market as oral remedies for diabetes patients, including acarbose, miglitol, voglibose, and metformin, among others. The long term use of these synthetic *α*-glucosidase inhibitors has been reported to cause multiple detrimental effects including liver disorders, abdominal pain, hepatic injury, abdominal fullness, flatulence, and diarrhoea [5]. According to the researchers, antioxidants play an important role in preventing complications of diabetes and recovering insulin sensitivity by protecting the *β*-cell against apoptosis that occurred during oxidative stress [6]. Therefore, the antioxidant and *α*-glucosidase inhibitor-rich traditional medicinal plant-based products may be an excellent option for treating or controlling diabetes without any adverse side effects.

In the current research, the fruit of *P. macrocarpa* were chosen to determine whether they could tackle the diabetic problems owing to the traditional claims for its use in the management of diabetes. The fruit of this traditional medicinal plant are widely used in Malaysia and neighbouring countries e.g., Indonesia, Philippines, Thailand, etc. to treat diabetes, kidney, cancer, impotency, heart disease, etc. [7]. Different researchers have earlier reported an antihyperglycaemic effect with different doses of conventional extracts of fruit of *P. macrocarpa*, viz., methanol extract 1000 mg/kg bw, *n*-butanol extract 1000 mg/kg, different sub-fractions of methanol extract 1000 mg/kg bw, water extract 1000 mg/kg bw, petroleum ether 1000 mg/kg bw, and ethyl acetate extract 500 mg/kg bw [8,9,10]. Despite the extensive use of this plant by indigenous people to treat various ailments, only conventionally prepared extracts using hazardous organic solvents have so far been examined to confirm its traditional claims in the management of diabetes without determining *α*-glucosidase inhibitors [11]. Furthermore, *α*-glucosidase inhibitors have not yet been identified from the extracts prepared through non-conventional extraction techniques that overcome the disadvantages of conventional extracts that require a huge amount of toxic solvent, a prolonged time, and are associated with breakdown of the thermolabile compounds, overheating, and a great chance for the toxic solvent to remain in the final product.

In vitro antioxidants and *α*-glucosidase inhibitions in plant extracts are considered important because they have been linked to antihyperglycaemic action via the inhibition of glycation of proteins, the deactivation of different enzymes, and changes in the collagen basement membrane of pancreatic *β*-cells [12]. In this regard, numerous antioxidants and *α*-glucosidase inhibition assays are used to determine the antioxidant activity of plant extracts; however, the most often employed technique is a colorimetric approach that is based on the scavenging of the DPPH (2,2-diphenyl-1-picrylhydrazyl) radical. The assay’s concept reflects the reducing process in which the acceptance of a hydrogen (H^.^) radical atom from the scavenger molecule, i.e., antioxidant, resulting in the reduction of DPPH to DPPH_2_, as seen when the purple colour changes to yellow [13]. As a result, the DPPH assay seems to be simple in nature, yet it is fast and reliable owing to its stable radical [14]. On the other hand, the *α*-glucosidase inhibitory test is employed to evaluate the *α*-glucosidase inhibitory action of plant extracts by measuring the release of *p*-nitrophenol from *p*-nitrophenyl *α*-*D*-glucopyranoside (PNPG). Computational techniques, viz., molecular docking and molecular dynamics simulation, are important sophisticated tools that are used to predict protein-ligand interactions and their mechanism as well [15]. Molecular dynamic simulation (MDS) is a technique for evaluating the physical motions of atoms, molecules and proteins which is executed on a wide range of protein systems and consequently, provides important insights on protein stability, protein-ligand binding, and protein-protein association, etc. [16].

Therefore, the initial goal of this study was to determine the in vitro antioxidant and *α*-glucosidase inhibitory potentials of all the extracts from fruit of *P. macrocarpa* prepared through conventional and non-conventional extraction techniques. Subsequently, the most potent extract was analysed with Quadrupole Time-of-Flight Liquid Chromatography Mass Spectrometry (Q-ToF-LCMS) analysis to identify *α*-glucosidase inhibitors. In order to clarify the molecular mechanism of the ligand-protein interaction, a computational investigation using in silico molecular docking, ADME/T parameters, and molecular dynamic simulations was conducted to confirm their antidiabetic potential.

## 2. Materials and Methods

### 2.1. Plant Material Collection and Identification

Fresh ripe red fruit of *P. macrocarpa* were collected from the nursery at Guar Perahu, Bukit Mertajam, Pulau Pinang and identified (voucher no. IIUM/308/15/2/1/NMPC19-1/13) by Dr. Norazian Binti Mohd. Hassan (Botanist), Kulliyyah of Pharmacy, International Islamic University Malaysia. Fruit of *P. macrocarpa* contain white, fibrous, and watery flesh. Unripe fruit are green in colour and they turn red when fully ripe. Fruit vary with the size of the plant. About 3–4 kg fresh and fully ripened fruit were obtained from one tree having a height of 5–6 m. The shape of the fruit was an eclipse, and the diameter was about 3 cm; they contained 1–2 seeds. Twenty kilograms of ripened fruit were collected in total from six different trees located at the botanical garden in Pinang, Malaysia. The weight of each fruit was about 60–70 gm. After separating the seeds, 17.6 kg of fruit flesh and 2.4 kg of seeds were obtained. Subsequently, the fruit flesh was dried at room temperature in the lab dryer for a week to yield 1.8 kg dry fruit flesh material, which was pulverized to a coarsely powdered form using a FRITSCH PULVERISETTE 19 Universal Cutting Mill (Germany), and the ground fruit flesh material was used for the different extraction processes [10].

### 2.2. Extraction Process

#### 2.2.1. Subcritical Carbon Dioxide Extract (SCE) Preparation

The subcritical carbon dioxide extraction technique described by Easmin et al. (2017) was followed, with some adjustments [17]. Briefly, fruit powder (200 gm) of *P. macrocarpa* was placed in an extraction vessel and soaked in ethanol at two distinct ratios, viz., sample: ethanol at 1:1 (SCE-1) and 1:2 (SCE-2). The proportion of feed to carbon dioxide solvent was 24:1. The feed material was extracted at a pressure of 7.0 MPa, a temperature of 28.7 °C, 200 cycles, and a run duration of 600 min. The extraction was then initiated by the entry of subcritical carbon dioxide into the vessel. The extract was distinguished from the subcritical CO_2_ by the process of vaporization that took place in the reboiler. The extracted material was then evaporated in a rotary evaporator and frozen at −80 °C until further analysis.

#### 2.2.2. Heating under Reflux Extract (HRE) Preparation

The heating under reflux extraction (HRE) of the sample of pulverized fruit of *P. macrocarpa* was accomplished using a slight modification of Pudziuvelyte et al.’s [18] method. Briefly, 30 gm ground fruit flesh powder was put in 95% ethanol and refluxed at 95 °C for 2 h followed by filtration using Whatman filter paper number 1. The residue leftover was again subjected to the same process (95% ethanol, 95 °C for 2 h) and repeated until no coloration was observed and was eventually evaporated by a rotary evaporator (IKA RV10). The same procedure was repeated three times following the same conditions.

### 2.3. Bioactivity Measurement

#### 2.3.1. DPPH Free Radical Scavenging Activity

The radical scavenging activity (DPPH) was measured using the technique described by Aryal et al. (2019), with minor modifications [19]. In this method, 1.972 mg of DPPH (0.1 mM) was freshly dissolved in 50 mL methanol before starting the work and put in a dark place. Different concentrations (2000–62.5 µg/mL) of test samples were prepared in the same solvent methanol. Initially, 80 μL of prepared DPPH (0.1 mM) was taken in a 96 microwell plate and then added 20 μL of different concentrations of the test extracts. The same amount of methanol was added to the control, ascorbic acid. Afterward, it was allowed to remain in the dark at room temperature for 10 min, following which the absorbance was taken at 517 nm. The percentage inhibition of the DPPH radical was obtained from the following Equation (1).
(%) DPPH = {(Ac − As)/Ac} × 100(1)
where Ac represents the control absorbance and As corresponds to the sample absorbance at 517 nm.

#### 2.3.2. α-Glucosidase Inhibitory Activity

The enzyme inhibitory test was conducted using Murugesu et al.’s (2019) standard procedures with minor changes [20]. In brief, dimethyl sulfoxide (DMSO)was used to dissolve the experimental samples (2–0.625 mg/mL) and the positive control (1 mg of quercetin). After that, the material was mixed with 15 µL/well of *α*-glucosidase from *Saccharomyces cerevisiae* enzyme (Megazyme, Ireland) that was prepared from a 50 mM buffer (pH 6.5). Then, 115 µL of 30 mM buffer (pH 6.5) was added to each well, and it was let to sit at 25 °C for 5 min. The reaction was then started by adding 75 µL/well of PNP *α*-glucoside and allowed to run for 15 min at room temperature. Finally, 50 µL of glycine with a pH of 10 was added to stop the reaction. The absorbance of each well (A) was recorded at 405 nm with a microplate reader (Tecan Infinite M200, Switzerland) to indicate the amount of *p*-nitrophenol released by PNP-*α*-glucoside. The *α*-glucosidase inhibitory activity (%) was calculated using Equation (2):(2)$$\%\;\mathrm{Inhibition}\;\mathrm{of}\;\mathrm{sample}\;=\frac{a_{n}-a_{s}}{a_{n}}\times 100$$
where a_n_ = Negative control, a_s_ = (Sample Absorbance-Blank Sample Absorbance).

### 2.4. Detection of Bioactive Compounds of SCE-2 by Quadrupole Time-of-Flight Liquid Chromatography Mass Spectrometry (Q-ToF-LCMS)

The bioactive compound analysis was carried out using the Agilent 1290 Infinity and 6520 iFunnel Q-ToF-LCMS (Agilent Technology, Santa Clara, Calif.) fitted with an electrospray interface and operating in positive ion mode, as reported by Saleem et al. (2019) with minor modifications [21]. About 20 µg of SCE-2 was made by dissolving it in 200 µL of methanol. The column (2.1 × 150 mm, 3.5 µm Agilent Zorbax Eclipse XDB-C18) was kept at 25 °C, while the auto-sampler was kept at 4 °C. Fresh 0.1% formic acid in water and 0.1% formic acid in acetonitrile mixtures were prepared for the mobile phase A and B, respectively. The flow rate at 0.5 mL/min, injection volume at 1.0 μL, the run time was 25 min, and the recovery period was 5 min. Using an electrospray ion source in positive mode, full scan MS analysis was done over the m/z 100–1000 range. The experiment was conducted using a capillary voltage of 3500 V. The data were processed using Agilent Mass Hunter Qualitative Analysis B.05.00 (Method: Metabolomics-2017-00004.m). The compounds were discovered by comparisons and searching the METLIN database.

### 2.5. Pharmacokinetic and Toxicity Studies

The SMILES codes of the Q-ToF-LCMS identified compounds were collected from the Pub-Chem database. The physiochemical properties of the identified compounds were determined through the pkCSM online database (https://biosig.lab.uq.edu.au/pkcsm/prediction; accessed on 12 January 2022). Furthermore, SwissADME (http://www.swissadme.ch/; accessed on 13 January 2022) was also used to identify “Drug-likeness” and pharmacokinetic properties of all the identified compounds. The LD_50_ and toxicity class were obtained through the Pro Tox-II database (https://tox-new.charite.de/protox_II/; accessed on 13 January 2022) [22].

### 2.6. In Silico Molecular Docking Analysis

Molecular docking was performed using AutoDock Vina (version 1.1.2) to evaluate the predictive binding affinity of the putatively identified bioactive compounds to the active site of the *α*-glucosidase (AG) enzyme. The crystallographic structure of the AG enzyme (PDB code: 3A4A) was collected from a protein data bank (PDB) https://www.rcsb.org/ (accessed on 20 January 2022) in .pbb format [23]. The three-dimensional (3D) structure of quercetin (positive control) and identified compounds through Q-ToF-LCMS analysis were collected from the PubChem database (https://pubchem.ncbi.nlm.nih.gov/; accessed on 21 January 2022) in .sdf format. All compounds and positive control (quercetin) were converted to .pdb format using Chimera (version1.15) software. AutoDock tool (version 1.5.6) was used to add Gasteiger charge and saved in .pdbqt format prior to molecular docking. The grid box was designed at 19.34, −0.74, and 22.04 while dimensions were 22 Å, 26 Å, and 20 Å for Z, Y, and X, respectively. The software PyMOL (PyMOL2 version 2.4.1) was employed to exhibit the initial visualization of 3D superimposed diagram of the 3A4A-compounds and control interactions. Finally, the combined files were saved in .pdb format. Later, these individual combined .pdb files were investigated to exhibit/confirm the number of hydrophilic and hydrophobic interactions, amino acid residues and bond distance through Biovia Discovery Studio Visualizer (San Diego, CA, USA) [24].

### 2.7. Molecular Dynamic Simulation (MDS) Approach

The molecular dynamic simulation (MDS) was performed in a Linux environment using the “Desmond v3.6 Program” in Schrödinger (https://www.schrödinger.com/; accessed on 10 Febuary 2022) (Paid version) to determine the thermodynamic stability of receptor-compound complex structures [25]. The *α*-glucosidase (3A4A)-ligand complex structures were investigated using 100 ns MDS to evaluate the binding consistency of the six compounds, namely lupenone (CID: 92158), swertianolin (CID: 5281662), *m*-coumaric acid (CID: 637541), pantothenic acid (CID: 6613), phytosphingosine (CID: 122121), and 8-*C*-glucopyranosyleriodictylol (CID: 42607963) to the targeted protein, *α*-glucosidase (PDB ID: 3A4A) [26].

For this context, a pre-determined TIP3P water approach was designed to ensure a specified volume with the orthorhombic periodic bounding box form at a distance of 10 Å. The framework was given an electrically neutral state by selecting suitable ions, such as O^+^ and 0.15 salt, and then randomly inserting them into the solvent solution. Following the construction of the solvency protein system with the compound complex, the system framework was reduced and loosened using the standard technique done using force field parameters OPLS3e inside the Desmond module and the standard procedures. The NPT assemblies that used the Nose-Hoover temperature combination and the isotropic method were held at one atmospheric pressure (1.01325 bar) and 300 K with 50 PS capture periods with a total energy of 1.2 kcal/mole. The MDS screenshots of all compounds were produced using Schrödinger’s maestro programme v9.5. The trajectory performance was used to assess the sustainability of the protein-ligand complex structure using root mean square deviation (RMSD), radius of gyration (Rg) value, root mean square fluctuation (RMSF), molecular surface area (MolSA), solvent accessible surface area (SASA) and polar surface area (PSA).

### 2.8. Compounds Target Pathway and Data Set Enrichment Analysis

Target genes were extracted using the STRING database using protein and the pathway was derived using the David database of the *α*-glucosidase enzyme. Later, drug compounds-enzyme, enzyme-target genes and target genes-pathway were merged in cystoscope (version 3_9_1) to show the compound target pathway analysis. Subsequently, the molecular function, cellular component, biological process and KEGG pathway of the identified genes were also determined through the https://david-d.ncifcrf.gov/list.jsp; (accessed on 24 June 2022) database [27].

### 2.9. Statical Analysis

GraphPad prism (version 7.00) software was used to assess the data for statistical differences using one-way ANOVA and two-way ANOVA, followed by Tukey’s and Dunnet’s multiple comparison tests. Different software and analysis processes are also mentioned in the text where necessary.

## 3. Results

### 3.1. DPPH Free Radical Scavenging Activity

The DPPH scavenging properties of plant extracts were investigated at six different concentrations (400, 200, 100, 50, 25, and 12.5 µg/mL) and the percentage of inhibitions at each concentration is shown in Table 1. The SCE-2 sample was found to exhibit the highest radical scavenging activity (75.36 ± 0.82%) among all the tested concentrations of extracts (SCE-1 and HRE) when compared to the standard, ascorbic acid.

### 3.2. Enzyme Inhibition Assay (α-Glucosidase)

The inhibitory effect of the tested samples along with the control (quercetin) was analysed at six different assay concentrations, viz., 80.0, 40.0, 20.0, 10.0, 5.0, 2.5, and 1.25 µg/mL (Table 2). The highest inhibitory effect of respective SCE-2, SCE-1, and HRE extracts was measured as 81.79 ± 0.82%, 73.21 ± 0.31%, and 67.57 ± 0.68% inhibitions, respectively. The positive control, quercetin, exerted the highest α-glucosidase inhibitory activity with 91.53 ± 0.24% inhibition.

### 3.3. Quadrupole Time-of-Flight Liquid Chromatography Mass Spectrometry (Q-ToF-LCMS) Analysis

The Q-ToF-LCMS technique was used to identify the bioactive secondary metabolite components of the SCE-2 from fruit of *P. macrocarpa*. As shown in Figure 1, a representative chromatogram of the SCE-2 with mass spectrometric detection in positive ion mode revealed complicated patterns of peaks. The tentative identified compounds of the SCE-2 are shown in Table 3 and their structures drawn using ChemDraw Ultra 12.0 software are shown in Figure 2.

### 3.4. Physiochemical, Pharmacokinetic and Toxicity Studies

The physiochemical and pharmacokinetics findings of all identified compounds are shown in Table 4. According to the PKCSM and SwissADME analyses, all compounds followed Lipinski’s drug rules without violation except compounds **12** and compound **13** that had two violations of Lipinski’s drug rules. In pharmacokinetics, all compounds also showed no hepatotoxic effects and except compounds **6**, **7**, and **11**, all compounds expressed no amex toxicity as well. The toxicity classes of all compounds were from **4** to **6** with different LD_50_ doses.

### 3.5. In Silico Study of Compounds Identified by Q-ToF-LCMS Analysis

According to the docking results, xestoaminol C showed the lowest binding affinity to the enzyme, whereas 8-*C*-glucopyranosyleriodictylol exhibited the highest binding affinity (lowest value) among the 14 identified docked compounds. In addition, except xestoaminol C, phytosphingosine, and C16 s phinganine, all compounds showed the higher binding affinity towards the 3A4A enzyme compared to the co-crystallized control ligand, *α*-*D*-glucose (ADG) (−6.0 kcal/mol). Furthermore, lupenone, 3-isomangostin hydrate, swertianolin, and 8-*C*-glucopyranosyleriodictylol exhibited the higher binding affinity when compared to that of quercetin (−8.4 kcal/mol). The binding affinity values of all the identified compounds are shown in Table 5.

Figure 3 shows a Pymol2-generated 3D overlay graphic that describes the simulated binding location of all 14 compounds, as well as ADG and quercetin, on the enzyme (3A4A). All molecules bound to the AG enzyme domain A, which includes all the catalytic residues, as indicated in the diagram. All the identified compounds had almost the same binding site as the ligand control (ADG) and standard (quercetin), indicating that they could all inhibit the enzyme in the same manner. The 14 identified compounds, ADG and quercetin are represented in Table 6 by bond distance, bond type and amino acid residues included in the binding interactions.

From the *m*-coumaric acid–α-glucosidase (3A4A) docked complex (Figure 4), three hydrogen bonds were built by TYR158 (2.54 Å), ASP215 (1.96 Å), and GLU277 (2.66 Å) amino acid residues with the hydroxyl moiety of *m*-coumaric acid. Among these, the ASP215 amino acid residue was predicted to be the strongest. Apart from this, three amino acid residues, namely ARG442 (3.69 Å), ASP352 (4.54 Å), and TYR72, formed Pi-Pi-T shaped with the aromatic moiety of this compound. Seven amino acid residues were involved in the pantothenic acid–α-glucosidase docked complex (Figure 4) in which hydrogen bonds were formed by ARG442 (2.66 Å) and HIS351 (1.94 Å) amino acid residues that interacted through the carbonyl moiety of pantothenic acid. The ARG213 (2.18 Å) and ASP215 (1.99 Å) amino acid residues interacted with the only hydroxyl moiety. In contrast, the GLU277 (2.66 & 2.19 Å) and ASP352 (2.16 & 2.99 Å) amino acid residues established with both hydroxyl moiety and the amino moiety of the pantothenic acid. Only one amino acid residue, viz., PHE303, built a Pi-alkyl interaction with the aliphatic moiety of the pantothenic acid at distance of 4.87 Å. In the xestoaminol C-3A4A docked complex (Figure 4), two hydrogen bonds interactions were formed by the PRO312 (2.70 Å) and SER240 (2.45 Å) amino acid residues with the hydroxyl moiety and amino moiety of xestoaminol C, respectively, while TYR158 (5.0 and 4.97 Å) and PHE303 (4.84 and 5.10 Å) amino acid residues built a Pi-alkyl interaction via the aliphatic moiety of this compound. From the 2,3,4′-trihydroxy-4-methoxybenzophenone-3A4A complex, three hydrogen bonds were formed by ASP233 (2.68 Å) and ASN415 (2.53 Å) amino acid residues with the hydroxyl moiety and LYS156 (2.85 Å) amino acid residue with the carbonyl moiety of this compound. While GLU429 (3.62 Å) and ASN317 (3.06 and 3.58 Å) amino acid residues formed a carbon-hydrogen bond, the HIS423 amino acid residue built a Pi-Pi-T-shaped interaction via aromatic moiety at a bond distance of 5.16 Å. An Alkyl interaction was formed by the LYS432 amino acid residue with the aliphatic moiety at a distance 3.99 Å. In addition, LYS432 (4.87 Å) and ALA418 (4.19 Å) amino acid residues formed a Pi-alkyl interaction via the aromatic moiety of this ligand (Figure 4).

In the C16 sphinganine and 3A4A docked result, three hydrogen bonds were formed via the hydroxyl moiety of this compound with GLN353 (2.50 Å), ASP352 (2.19 Å), and GLU411 (2.46 Å) amino acid residues. Furthermore, LYS156 (5.02 Å) and ARG315 (4.62 Å) amino acid residues established an alkyl interaction while PHE303 (5.43 Å) and TYR158 (3.98 Å) amino acid residues interacted through a Pi-alkyl interaction with the aliphatic moiety of the C16 sphinganine (Figure 4). From the swertianin-3A4A docked result (Figure 4), one hydrogen bond interaction was formed by the ARG315 amino acid residue at a bond distance of 2.05 Å, while one carbon-hydrogen bond was built via the ASP352 amino acid residue at a bonding distance of 3.59 Å. Furthermore, the ARG315 (4.52 and 4.89 Å) and TYR158 (4.95 and 5.52 Å) amino acid residues interacted with the aromatic moiety of swertianin via Pi-alkyl and Pi-Pi-T-shaped interactions, respectively. In the emmotin A docking results (Figure 4), it was noticed that TYR158 amino acid residue interacted with the hydroxyl moiety of emmotin A via a hydrogen bond at a bond distance of 2.40 Å. Whereas, the ARG315 (4.38 Å) amino acid residue interacted with the aromatic moiety and the PHE178 and PHE303 amino acid residues associated with the aliphatic moiety through a Pi-alkyl interaction. From the docking results (Figure 4), it was found that in the docked complex containing phytosphingosine–α-glucosidase, the HIS280 (2.72 Å) and ASP307 (2.19 and 2.45 Å) amino acid residues were associated with the hydroxyl moiety, and the SER311 (2.46 Å) and PRO312 (2.20 Å) amino acid residues were interacted with the amino moiety of phytosphingosine through hydrogen bond interactions. In addition, the ARG315 amino acid residue made an alkyl interaction with the aliphatic moiety at a bond distance of 4.81 Å and 4.84 Å, respectively, whereas the TYR158 (4.78 and 5.41 Å), PHE303 (5.03 Å), PHE314 (5.48 Å), and HIS280 (5.47 Å) amino acid residues interacted with the aliphatic moiety of phytosphingosine via Pi-alkyl interaction.

From the 1-monopalmitin–α-glucosidase docking result (Figure 4), three hydrogen bonds were formed by the ARG442 (2.38 Å and 5.98 Å), ASP69 (2.50 Å), and ASP352 (2.52 Å) amino acid residues. In addition, the ASP352 (3.55 and 3.61 Å) amino acid residue interacted with aliphatic moiety through carbon-hydrogen bond interaction. Apart from this, three Pi-alkyl interactions were built by the ASP303 (5.08 Å), PHE314 (5.46 Å), and TYR158 (4.58 Å, 5.18 Å) amino acid residues, while two alkyl interactions were formed by the ARG315 (4.11, 5.06 Å) and LYS156 (4.20 Å) amino acid residues. Besides, the TYR158 amino acid residue interacted with the aliphatic moiety of this ligand via Pi-sigma interaction at a bond distance of 3.60 Å. There was no hydrogen bond formed in the lupenone-3A4A docked complex. Four alkyl interactions were formed between the aliphatic moiety of lupenone and four amino acid residues namely VAL308, ILE328, ALA329, and PRO312 at a bond distance of 5.29 Å, 5.29 Å, 3.59 Å, and 4.55 Å, respectively (Figure 4). In the case of the 3-isomangostin hydrate–3A4A docking complex (Figure 4), the result indicated that two hydrogen bonds were observed in the GLU332 (2.54 Å) and SER304 (2.16 Å) amino acid residues, while the HIS280 (2.66 Å) amino acid residue formed a Pi-donor hydrogen bond with the hydroxyl moiety of this ligand. Furthermore, the HIS280 amino acid residue also built a Pi-alkyl interaction with the aromatic moiety at a bond distance of 5.29 Å. On the other hand, the ARG315 (4.33 and 4.69 Å) amino acid residue was associated with an aliphatic moiety via alkyl interactions. In addition, the PRO312 (3.69 Å) and ASP307 (3.79 and 3.84 Å) amino acid residues interacted with the aromatic moiety of 3-isomangostin hydrate via Pi-sigma interaction and Pi-anion interaction, respectively. From the α-glucosidase and swertianolin binding interaction (Figure 4), two hydrogen bonds were formed with ARG442 and GLU411 amino acid residues at a bond distance of 2.92 Å and 2.35 Å, respectively. Furthermore, the TYR158 (5.50 and 5.53 Å) amino acid residue interacted with an aromatic moiety via Pi-Pi-T-shaped interactions while the ARG315 amino acid residue associated with the aromatic moiety via Pi-alkyl interaction at a bond distance of 4.74 Å. In the docked complex, 8-*C*-glucopyranosyleriodictylol (Figure 4) built interactions with the GLU277 (2.78 Å), ASP242 (1.89 Å), and SER240 (2.36 Å) amino acid residues via hydrogen bonds due to the presence of hydroxyl groups. On the other hand, the SER240 amino acid residue interacted with an aliphatic moiety through carbon-hydrogen bond interaction at a bond distance of 3.38 Å. In addition, one Pi-donor hydrogen bond was observed between the HIS280 amino acid residue and the aromatic moiety of this ligand at a bond distance of 3.21 Å. Furthermore, another amino acid residue, TYR158, built a Pi-Pi-T-shaped interaction with aromatic moiety at bonding distance of 4.99 Å. Among these amino acid residues, the ASP242 amino acid residue was predicted to be the strongest due to the hydrogen bonding interaction and bonding distance. According to Figure 4, the GLN353 (2.91 Å) and GLU411 (3.55 Å) amino acid residues were interacted with the hydroxyl moiety of longispinogenin via a hydrogen bond and carbon-hydrogen bond interactions, respectively. On the other hand, one alkyl interaction was also observed with VAL216 at a bonding distance of 4.76 Å. In addition, two Pi-alkyl interactions were formed by the PHE303 (5.36 Å) and TYR158 (5.41, 4.59 and 4.64 Å) amino acid residues. Among these amino acid residues in the longispinogenin-3A4A docked complex, GLN353 was predicted to exert the highest enzyme inhibitory activity. In the 3A4A-quercetin docked complex (Figure 4), it was found that the three hydrogen bonds were formed with the ARG315, GLH277, and ASH215 amino acid residues at a bond distance of 2.73, 2.01, and 2.46 Å, respectively. In addition, the ARG442 (3.75 Å), ASP352 (4.28 Å), and PHE303 (4.96, 5.13 Å) amino acid residues were interacted via Pi-cation, Pi-anion, and Pi-Pi T-shaped interactions, respectively, in the docked complex involving quercetin. From the ADG-3A4A docked complex result (Figure 4), it was noticed that six amino acid residues including ARG213, GLH277, ASH215, ASP352, HIE351, and ARG442 constructed hydrogen bond interactions with the control ligand at a bond distance of 1.80, 2.13, 2.44, 2.18 and 2.37, 1.96, and 1.90 Å, respectively.

### 3.6. Molecular Dynamic Simulation (MDS)

The average RMSD values of the six compounds, viz., lupenone (CID: 92158), swertianolin (CID: 5281662), *m*-coumaric acid (CID: 637541), pantothenic acid (CID: 6613), phytosphingosine (CID: 122121), and 8-*C*-glucopyranosyleriodictylol (CID: 42607963), were 1.047, 0.809, 0.994, 0.886, 0.782, and 0.961, respectively (Figure 5). The compounds’ RMSD (Å) values exhibited very little fluctuation, indicating that the protein-compound complex was conformationally stable in its structure.

The RMSF (Å) result of the specific compound—the 3A4A complex—was evaluated for 587 amino acid residues with little fluctuation (Figure 6). The six compounds, viz., lupenone (CID: 92158), swertianolin (CID: 5281662), *m*-coumaric acid (CID: 637541), pantothenic acid (CID: 6613), phytosphingosine (CID: 122121), and 8-C-glucopyranosyleriodictylol (CID: 42607963), exhibited RMSF values ranging 0.386–5.586 Å, 0.502–6.412 Å, 0.372–4.943 Å, 0.371–5.932 Å, 0.365–5.125 Å, and 0.375–5.471 Å, respectively. The most consolidated secondary structural elements, viz., α-helices and β-strands, were found to have the lowest measurement frequency in the distance of 5–225, 236–417, 427–556 amino acid residues.

The molecular surface area (MolSA) was calculated using a probe radius of 1.4 Å, which is equivalent to the van der Waals surface area of a water molecule. According to Figure 7, all the compounds, viz., lupenone (CID: 92158), swertianolin (CID: 5281662), *m*-coumaric acid (CID: 637541), pantothenic acid (CID: 6613), phytosphingosine (CID: 122121), and 8-C-glucopyranosyleriodictylol (CID: 42607963), in association with the targeted protein α-glucosidase (3A4A) acquired the standard van der Waals surface area with no fluctuation until 100 ns.

The Rg (radius of gyration) of an enzyme–compound interconnection network can be depicted as the set of its atoms across its axis. In the molecular dynamic simulation study, the highest fluctuation of the radius of gyration (Rg) values was observed in the docked complex containing phytosphingosine (CID: 122121) varying in the range between 3.661 and 6.882 nm. Whereas, the Rg of other ligands, viz., lupenone (CID: 92158), swertianolin (CID: 5281662), *m*-coumaric acid (CID: 637541), pantothenic acid (CID: 6613), and 8-*C*-glucopyranosyleriodictylol (CID: 42607963), exhibited (4.156–4.335 nm), (4.218–4.634 nm), (2.897–3.110 nm), (2.876–3.443 nm), and (4.081–4.420 nm), respectively, demonstrating that the protein’s binding site was not structural changed when the specified ligand molecules were bound Figure 8.

Another vital parameter for understanding the ligand’s solvent behaviour is solvent-accessible surface area (SASA). The SASA result of the six ligands was exhibited between 0 and 666.249 Å^2^. Among the six compounds, lupenone, *m*-coumaric acid, and phytosphingosine were found to exhibit higher values with fluctuations (Figure 9). Furthermore, the PSA of the selected six compounds value ranged between 33.363 Å^2^ and 384.646 Å^2^. All compounds found a potent value with the α-glucosidase protein without fluctuation (Figure 10).

### 3.7. Compound-Target-Pathway Network

The 11 vital genes involved in the α-glucosidase protein interaction network were generated from the STRING database (Figure 11). The 14 compounds, 11 gene targets, and the top 8 pathways were imported into Cytoscape_v3.9.1 software, and the “compound-target-pathway” network was obtained as shown in Figure 12. The pink colour represents the chosen drugs, a red colour represents the enzyme receptor, a green colour indicates the genes that are involved in the α-glucosidase inhibition pathway as shown in the cyan colour.

### 3.8. Gene Set Enrichment Analysis

Enrichment analysis is a flexible technique for learning about the pathways whose activity is impacted by a certain gene group. According to Figure 13A, the top 10 molecular functions that are enriched include hydrolase activity, hydrolysing O-glycosyl, α-1,4-glucosidase activity, α-glucosidase activity, α-amylase activity, amylase activity, galactosidase activity, glucosidase activity, β-galactosidase activity, protein homodimerization activity, chloride ion binding. The cellular elements shown in Figure 13B include the lysosomal lumen, azurophil granule, ficolin-1-rich granule, vacuolar lumen, ficolin-1-rich granule membrane, secretory granule lumen, lysosome, and tertiary granule membrane. Our biological process enrichment analysis revealed, as shown in Figure 13C, that the list of genes targeted by the plant compounds was significantly related to N-glycan processing, glycosphingolipid metabolism, glycolipid metabolism, glycoprotein metabolism, sphingolipid metabolism, neutrophil degranulation, neutrophil mediated immunity, neutrophil activation involved in immune response, glycoside metabolism, and glycosylation. As shown in Figure 13D, the KEGG pathway annotation revealed that galactose metabolism, starch and sucrose metabolism, carbohydrate digestion and absorption, lysosome pathway, glycosphingolipid metabolism, salivary secretion, pancreatic secretion, and protein processing in endoplasmic reticulum were at the foremost on the list.

## 4. Discussion

### 4.1. In Vitro Bioactivity Assay

Our findings are based on the fruit extract of *P. macrocarpa* that was derived from an improved subcritical carbon dioxide method to assess the bioactive properties of the extract more accurately. Several parameters were considered, such as temperature, pressure, solvent properties, solvent to sample ratio, and extraction time. Another advantage of this method is the use of a low temperature (28.7 °C) that was maintained throughout the experiment to avoid the thermolabile breakdown of bioactive substances further ensuring the preservation of most of the bioactive compounds present in the resultant extract [28]. In addition, a pressure of 70 bar was also maintained throughout the extraction process. This process is not only considered cost-effective and energy saving but also helps to preserve the bioactive compounds and their physical properties. Furthermore, food-grade ethanol was used in the extraction process; this is a polar solvent and helps to extract polar compounds (i.e., flavonoids, phenolic acids, saponins etc.) from the plant material. It also helps to avoid any undesirable contamination from the toxic solvents, consequently, no toxic solvent is left in the final product, thereby further ensuring the safety of the extract obtained through the subcritical carbon dioxide extraction method [29].

In this research, we analysed a conventional extract (HRE) and non-conventional extracts (SCE-1 and SCE-2) for a better understanding of the in vitro antioxidant and *α*-glucosidase inhibitory effects of these extracts. Both antioxidant and *α*-glucosidase inhibitions are involved in antidiabetic action through the inhibition of different enzymes and pathways [12]. The key mechanism of DPPH scavenging free radicals is the transfer of hydrogen atoms in the form of radicals. Many investigations have demonstrated that the scavenging mechanism of phenolic or other types of similar compounds could easily transfer a hydrogen atom as a radical to a free radical. In this research, the radical scavenging ability of the *P. macrocarpa* extract was found to be in the order as SCE-2 > HRE > SCE-1. In this research, SCE-2 exhibited higher DPPH activity (75.36 ± 0.82%) compared to the previous study reported by Soeksmanto et al. (2007) in which methanol and ethanol extracts of fruit of *P. macrocarpa*, prepared through a conventional extraction method, were able to produce antioxidant activity of 38.4 and 48.1%, respectively [30]. The manifestation of a higher DPPH activity exerted by SCE-2 in this research study could be easily attributed to the non-conventional extraction technique as well as to the food-grade ethanol, which was used as an extraction solvent because it has been earlier reported to improve the solvating power of carbon dioxide and the yield of flavonoid/polyphenol-rich extract [31]. Furthermore, the polarity of the solvents influences antioxidant activity owing to the differences in various polar molecules in the extracts [32].

Inhibition of the *α*-glucosidase enzyme is one of the effective methods for managing carbohydrate metabolic disorders, such as T2DM. The *α*-glucosidase inhibitory efficacy of the *P. macrocarpa* fruit extracts was in the order of SCE-2 (81.79 ± 0.82%) > SCE-1 (73.21 ± 0.31%) > HR (67.57 ± 0.68%) at the highest concentration (80 µg/mL) that was found higher compared to the results reported by Ali et al. (2013) in which butanol and methanolic extracts at 100 μg/mL showed 75% and 32% *α*-glucosidase inhibitory effect, respectively [33]. This result may also be attributed to the concentrations of extracted biologically active components and their fluctuation at various solvent ratios. When the ethanol content of the SCE-2 extract was raised, it demonstrated very strong inhibitory action against *α*-glucosidase, which was substantially different (*p* < 0.05) than the *α*-glucosidase inhibitory activity of SCE-1 and HRE. The recovery of the more lipophilic as well as hydrophilic compounds was higher at higher ethanol concentrations, owing to the destruction of cell membranes and penetration that allowed for more effective extraction [34]. The fact that SCE-2 had the most potent action in this investigation indicates that more lipophilic compounds along with the hydrophilic compounds in *P. macrocarpa* fruit may have been responsible for *α*-glucosidase inhibitory activity. This finding was found to be consistent with the previous findings that absolute organic solvents containing subcritical carbon dioxide had more biological activity than a combination of aqueous and organic solvents [17]. The phenolic and flavonoid contents have a critical role in determining the inhibitory action of *α*-glucosidase [35]. The current investigation discovered that the SCE-2 had the higher levels of phenolic and flavonoid contents, which may explain its potent and persistent *α*-glucosidase inhibitory effect. In general, the SCE-2 had the highest antioxidant and antidiabetic activities of the extraction procedures owing to the solvent ratio, optimal temperature and pressure, and availability of many bioactive metabolites. As a result, SCE-2 derived *α*-glucosidase inhibitors may be more preferred and safer natural products for diabetes treatment and prevention.

### 4.2. Qualitative Tandem Liquid Chromatography Quadrupole Time-of-Flight Mass Spectrometry (Q-ToF-LCMS) Analysis

Overall, the Q-ToF-LCMS results showed that the SCE-2 of *P. macrocarpa* contained different classes of natural products in the form of flavonoids, fatty acids, and other phenolic compounds. Among the compounds identified through Q-ToF-LCMS analysis, most of the compounds have also been identified or isolated from other plants and have earlier been reported to exhibit antidiabetic and antioxidant activities through different in vitro and in vivo assays. Among the most important identified compounds, *m*-coumaric acid has been reported to play an important role in diabetes and lipid metabolism [36,37]. Moselhy et al. (2018) reported that 150 mg/kg bw of *m*-coumaric acid reduced the fasting blood sugar and HbA1c level after six weeks administration in rats [36]. Hazelwood (1956) reported pantothenic acid as an insulin sensitiser [38]. Xestoaminol C and C16 sphinganine were earlier identified from *Marantodes pumilum* extract using LCMS and found to exert antidiabetic effect in the form of decreasing insulin resistance via up-regulating PPAR-γ [39]. 2,3,4′-trihydroxy-4-methoxybenzophenone isolated from fruit extract of *P. macrocarpa* was found to reduce the cholesterol level in the high-fat diet-induced diabetic rat [40,41]. Other compounds, namely C16 sphinganine and emmotin A, were also reported to be identified by different researchers in different plants crude extracts, displaying an effect as insulin inducers [42,43]. Swertianin has been reported to play a role as an antioxidant and antidiabetic agent [44,45]. Sphingosine has been reported to exert an effect on the metabolism process of obese in type-2 diabetes [46]. Phytosphingosine has been demonstrated to exhibit an effect on glucose intolerance, insulin sensitivity and decreasing cholesterol levels [47,48]. 1-monopalmitin has been identified from crude extract using GC-MS and was reported to exert a potential effect for the management of T2DM [49,50]. Moreover, 1-monopalmitin was also predicted to be an *α*-glucosidase inhibitor using molecular docking approach by Murugesu et al. (2018) [51]. Lupenone has been reported to play a role in type-2 diabetes by downregulation of PPARγ [52,53]. 3-Isomangostin hydrate found after optimization of plant material of *Garcinia mangostana* using a non-conventional supercritical carbon dioxide extraction technique was reported to exert an antioxidant effect [54]. 8-*C*-Glucopyranosyleriodictylol was found in an extract of *Monotheca buxifolia* showed antioxidant effects [55]. This is the first report on the preliminary Q-ToF-LCMS analysis of SCE-2. Based on Q-ToF-LCMS analysis of the bioactive compounds, it is noticed that fruit of *P. macrocarpa* has strong antidiabetic effects. *α*-glucosidase inhibitors retard the action of *α*-glucosidase on the hydrolysis of carbohydrates, thereby delaying the carbohydrate digestion from the small intestine at postprandial conditions as a result decrease the glucose level in type-2 diabetes patients which occurs due to the impairment of insulin sensitivity and pancreatic *β*-cell. Similarly, antioxidants have been reported to play a crucial role in restoring insulin sensitivity by protecting the *β*-cell against apoptosis that occurs during oxidative stress [1,6]. Hence, all the hit compounds were considered for an in silico molecular docking to determine their mechanism of action as *α*-glucosidase inhibitors.

### 4.3. Pharmacokinetics and Physiochemical Analysis

Physiochemical and pharmacokinetic properties are considered for the therapeutic effect and safe dose determination before discovering any drug. In overall findings, we noticed that all compounds belonged to a toxicity class between **4** and **6**, and no hepatoxicity was also observed. Furthermore, all compounds followed the five drug-likeliness rules except compounds **12**, **13**, and **14**, which were found to have under two violations. Fewer violations are considered more effective for drugs; that is why some violations can still be allowed for a compound to be accepted as a drug. Interestingly, some drugs, such as acarbose, violate three drug likeliness rules, but the latter is still used as antidiabetic drug [56]. That is why we considered all the hit compounds for in silico molecular docking to determine their mechanism of action.

### 4.4. Molecular Docking of Compounds Identified by Q-ToF-LCMS Analysis

Although it has been reported through different structure-–activity relationship studies that the presence of hydroxyl groups in a molecule is responsible for its *α*-glucosidase inhibitory action [57,58,59], the compounds found using Q-ToF-LCMS in this investigation were mostly polar in nature due to the presence of polar functional groups, particularly a hydroxyl group. These polar functional groups might have helped to generate the strong interactions between proteins and inhibitors, which result in the creation of likely conformational complexes, which may have resulted in a positive enzyme inhibitory impact of the compounds identified by Q-ToF-LCMS.

The binding affinity of xestoaminol C-AG (−5.5 kcal/mol) and phytosphingosine-AG (−5.8 kcal/mol) was found lower than ADG and quercetin. In both docked-complexes, no catalytic residue was found, which was similar to the ADG-AG docked complex. As there was no catalytic residue detected to be involved in the aforementioned docked complexes, these compounds might have bound in the allosteric site as non-competitive inhibitors towards the enzyme [51,60,61].

In the *m*-coumaric acid-AG docked complex, it was noticed that three catalytic residues, namely ARG442, ASP215, and GLU277 were found similar to the ADG-AG docked complex. Moreover, ASP215 and GLU277 were also found as catalytic residues in the AG enzyme. Seven amino acid residues involved in the pantothenic acid-AG docked complex, among them four residues (viz., ASP215, ASP352, GLU277, and ARG442) also showed interaction in the ADG–AG complex via a hydrogen bond. In the 2,3,4′-trihydroxy-4-methoxybenzophenone-AG docked complex, nine amino acid residues were found to be involved. Although three hydrogen bonds were found to be involved in this docked complex, there was no catalytic residue found to be involved similar to the ADG-AG docked complex. The binding affinity of C16 sphinganine was found to be like ADG but lower than quercetin. In the C16 sphinganine-ADG docked complex, there were seven amino acid residues found to be involved, among them three residues, viz., ASP352, GLN353, and GLU411, interacted via hydrogen bonds and four interacted through hydrophobic interactions. Among them, only one catalytic residue ASP352 was found, similar to the ADG-AG docked complex. Although there was only one catalytic residue found, other amino acid residues were also found to be involved via hydrophilic and hydrophobic interactions. In the swertianin-AG docked complex, there were also two catalytic residues involved like the ADG-AG docked complex. In this docked complex, fewer hydrogen bonds and fewer hydrophobic interactions were found than ADG-AG. Although amino acid residues were involved in one hydrogen bond and three hydrophobic interactions in the emmotin A-AG docked complex, there were no catalytic residues present in the docked complex. Therefore, the AG enzyme inhibition might have manifested by binding emmotin A in the allosteric site that alters the enzyme activity. In the 1-monopalmitin-AG docked complex, there were fewer hydrogen bonds and more hydrophobic bonds present than both AG and positive control quercetin. Due to more hydrophobic interactions, this compound stabilized the docked complex and enhanced the binding affinity of the ligand at the binding interface as a result influenced the carbohydrate digestion slowly [62]. Ten amino acid residues were involved in this docked complex, among them, only two catalytic residues were found similar to ADG-AG and quercetin-AG docked complexes. Five amino acid residues were present in the longispinogenin-AG docked complex, among them, there were no catalytic residues present like the control ligand docked. It may have been bound via an allosteric site due to the hydrophilic and hydrophobic interactions, although compound 2,3,4′-trihydroxy-4-methoxybenzophenone, 1 emmotin A, and longispinogenin have no catalytic residues but have more amino acid residues that interacted via hydrophobic interaction due to the presence of the extra methyl groups [63].

In the lupenone-ADG docked complex, there was no hydrogen bonding present, but four pi-alkyl interactions were found. Moreover, seven amino acid residues were involved in 3-isomangostin hydrate-AG, among them two amino acid residues interacted via a hydrogen bond and other residues were involved via hydrophobic interactions. There were also no catalytic residues present in this docked complex. In the swertianolin-AG docked complex, there were two hydrogen bonds and two hydrophobic bonds present. Among these hydrogen bonds, only one catalytic residue, ARG442, was involved like an ADG-AG docked complex. In the 8-*C*-glucopyranosyleriodictylol docked result, seven amino acid residues were involved in this complex, among these, one catalytic residue was present like the ADG-AG docked complex. The docked complexes of these compounds to the enzyme active site included a few hydrogen bonds. In order to generate interactive inhibition, the compounds’ interactions with the enzyme were mostly connected by hydrophobic contact. This was found to be consistent with the findings of [63]. Moreover, according to the docking data interpretation, all identified compounds had moderate to strong binding affinities to the enzyme’s active site, suggesting the capacity to bind, slow down the catalytic activity, and finally inhibit the enzyme.

### 4.5. Molecular Dynamic Simulation

Only six compounds, viz., lupenone (CID: 92158), swertianolin (CID: 5281662), *m*-coumaric acid (CID: 637541), pantothenic acid (CID: 6613), phytosphingosine (CID: 122121), and 8-*C*-glucopyranosyleriodictylol (CID: 42607963), were selected based on the binding affinity, nature of the compounds, toxicity, physiochemical and pharmacokinetic properties, probability of the *α*-glucosidase inhibitory effect, and previously reported data for the same compounds by other scholars. Among the six compounds, we selected phytosphingosine despite lower binding affinity than ADG and positive control to investigate the MDS findings to compare with other compounds which were found to demonstrate higher binding affinity than ADG. The RMSD measurement is the most fundamental indicator used to assess protein stability [64]. A RMSD value of less than 3 Å suggests that the system is acceptable and stable. The lower the value, the more stable the protein and vice versa [65]. Although the RMSD values of all the compounds showed an acceptable range, compounds pantothenic acid (CID: 6613), phytosphingosine (CID: 122121), and *m*-coumaric acid (CID: 637541), showed more stability due to little fluctuations than others.

Another critical indicator that reflects the stability of macromolecules is RMSF, which measures protein fluctuation [66]. The RMSF value was used to access the protein fluctuation, which also revealed less fluctuation, showing that the compound was more stable to the target protein. In our investigation, all the compounds showed less fluctuation and demonstrated a strong stability value in association with the selected 3A4A protein. Among the selected compounds, lupenone, swertianolin, phytosphingosine, and 8-*C*-glucopyranosyleriodictylol were found to demonstrate less stability than *m*-coumaric acid and pantothenic acid.

A protein’s stability is also linked to its MolSA; hence, a major change in the MolSA complex may result in instability, which is extremely undesirable [67]. After 100 nanoseconds (ns) of dynamic simulation, all compounds were found to be stable with no fluctuation. Among all six ligands, the compounds *m*-coumaric acid (CID: 637541) and pantothenic acid (CID: 6613) showed the lowest MolSA values and are considered for the highest stability and favourable in contrast to the other compounds (viz., lupenone, swertianolin, phytosphingosine, and 8-*C*-glucopyranosyleriodictylol).

The radius of gyration (Rg) helps to compute the centre of mass from C and N terminals of the protein that is under examination, the protein structure stability and provides a greater assessment of protein folding features [68]. The lower the Rg value, the greater the compactness, and the higher the Rg value, the greater the disassociation of the compound from the protein [69]. By comparing the other five ligands, phytosphingosine (CID: 122121) demonstrated the greatest gyration radius, suggesting that it was the most flexible and unstable among all the compounds. On the other hand, the ligands lupenone (CID: 92158), swertianolin (CID: 5281662), *m*-coumaric acid (CID: 637541), pantothenic acid (CID: 6613), and 8-*C*-glucopyranosyleriodictylol (CID: 42607963) exhibited stronger stability.

Biological protein structure and function are both affected by the amount of solvent-accessible surface area (SASA). Usually, amino acid residues on the surface of a protein act as active sites or interact with other ligands. This makes it easier to understand how a molecule behaves in a solvent (hydrophobic or hydrophilic) and how it interacts with ligands [70]. In our findings, protein structure was affected by solvent-accessible surface area due to the compounds lupenone, *m*-coumaric acid, and phytosphingosine, which might have occurred due to the absence of or fewer catalytic residues, hydrophilic and hydrophobic bonds, van der Waals radius as well as less binding affinity. The polar surface area (PSA) depends on the number of oxygen atoms present in the ligands. More PSA indicates the more oxygen atom present in the compounds [71]. In our finding, we found the highest PSA in compound 8-*C*-glucopyranosyleriodictylol due to the highest number of oxygen atom present. Similarly, the lowest PSA was found in lupenone due to the lowest number of oxygen atom present.

### 4.6. Compound Target Pathway and Gene Set Enrichment Analysis

The ligand/compound-target relationship elucidated that the antihyperglycemic treatment effectiveness of SCE-2 was precisely associated with the 14 compounds that were identified as possible components for treating diabetes in SCE-2. According to the findings of Moselhy et al. (2018), hyperglycaemia can be controlled by modulating the enzymes involved in pancreatic glucose metabolism [36]. According to the component-target–pathway network, the therapeutic impact of discovered compounds on diabetes interacted directly with 14 genes. The KEGG pathway enrichment study of *α*-glucosidase revealed that eight pathways were precisely linked to the onset and advancement of T2DM, proposing that all these pathways may be the molecular mechanism of SCE-2 against diabetes mellitus. To further confirm the gene ontology (GO), the molecular function (Figure 13A), cellular component (Figure 13B), biological process (Figure 13C), and KEGG pathway 2021 (Figure 13D) of the aforementioned 11 genes (Figure 11) were also found to slow carbohydrate digestion as well as show antidiabetic related activities through David database.

### 4.7. Limitation and Future Directions of the Study

Prior to orchestrating the in vivo and clinical studies, in vitro and in silico studies are generally considered crucial steps to know the fundamental and enzymatic effects of any drugs or supplement development. In our research, we initially evaluated an in vitro *α*-glucosidase inhibitory effect of the subcritical CO_2_ fruit extract of *P. macrocarpa*, which upon Q-ToF-LCMS analysis revealed some tentative *α*-glucosidase inhibitors. The in silico molecular dynamic simulations studies further confirmed the *α*-glucosidase inhibitory effect of all the putatively identified compounds present in subcritical CO_2_ fruit extract of *P. macrocarpa*. Nevertheless, these *α*-glucosidase inhibitors should be quantified to determine the exact amount of these compounds present in the extract. Furthermore, isolation of these compounds should be carried out to further confirm the presence of α-glucosidase inhibitors in the subcritical CO_2_ extract supporting it’s an in vitro *α*-glucosidase effect. Moreover, an in vivo research study is still required to further confirm the antidiabetic potential of the resultant extract. Animal models resembling humans, such as type 2 diabetes with a high fat diet, should be developed for in vivo studies and a comparative evaluation study is required on extract with the *α*-glucosidase inhibitors prior making any conclusion supporting its role as the herbal remedy to manage diabetes [72].

## 5. Conclusions

This study has confirmed the antihyperglycaemic potential of fruit of *P. macrocarpa* through in vitro and in silico *α*-glucosidase inhibition approaches. It was found that SCE-2 exhibited higher antioxidative and *α*-glucosidase inhibitory effects than HRE and SCE-1 extracts. In addition, the Q-ToF-LCMS analysis of the highest bioactive fraction (SCE-2) revealed fourteen metabolites responsible for the inhibitory activity of the *α*-glucosidase, which was further confirmed through an in silico molecular dynamic simulations approach. Based on the findings, six compounds, namely swertianolin, *m*-coumaric acid, pantothenic acid, phytosphingosine, and 8-*C*-glucopyranosyleriodictylol were further analysed for 100 ns duration to determine the stabilization and intermolecular relationships of a protein-compound complex. All compounds except phytosphingosine were found to have stronger stability, fewer fluctuations, and good compactness as confirmed by RMSD, RMSF, SASA, PSA, MolSA, and Rg analyses. Consequently, these ligands could be used as a potential biomarker for *α*-glucosidase inhibitor. Finally, it can be concluded that the SCE-2 possesses an antihyperglycaemic potential which was confirmed through in vitro, in silico MDS, and network pharmacology approaches for the first time through this research work.

## Figures and Tables

**Figure 1 metabolites-12-01267-f001:**
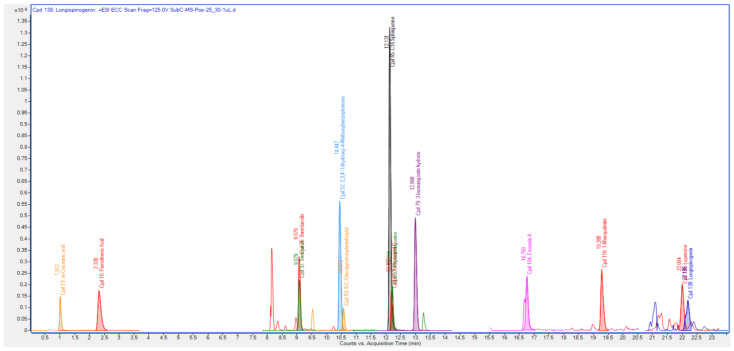
Chromatogram of Q-ToF-LCMS identified compounds.

**Figure 2 metabolites-12-01267-f002:**
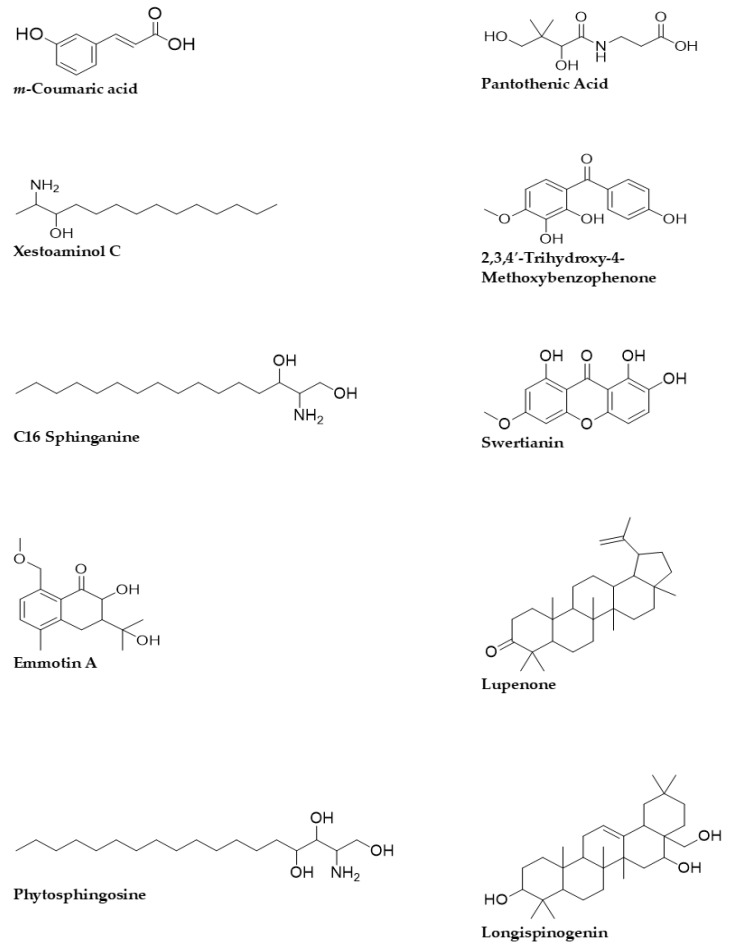
2D structures of the tentative compounds identified by Q-ToF-LCMS analysis.

**Figure 3 metabolites-12-01267-f003:**
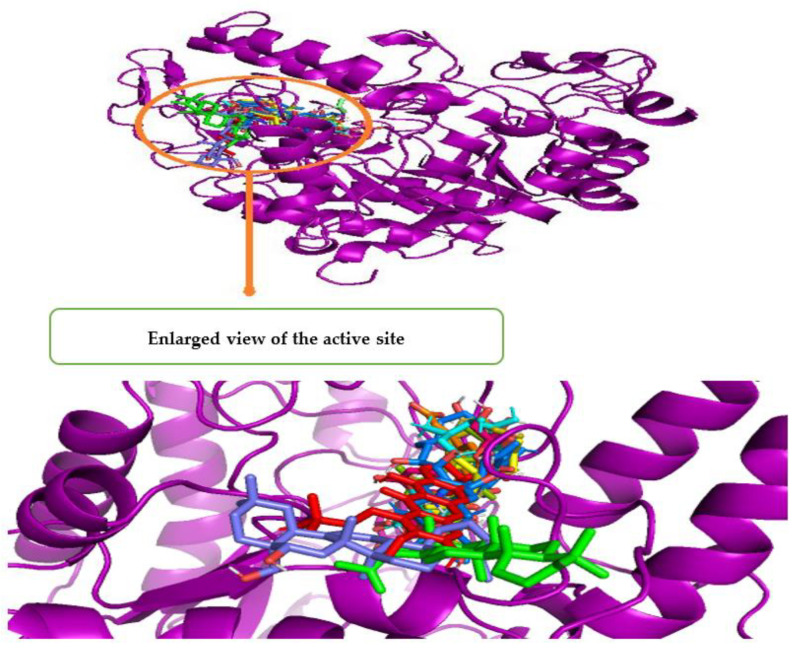
3D superimposed diagram of the bioactive compounds (*m*-coumaric acid, pantothenic acid, xestoaminol C, 2,3,4′-trihydroxy-4-methoxybenzophenone, C16 sphinganine, swertianin, emmotin A, phytosphingosine, 1-monopalmitin, lupenone, 3-lsomangostin hydrate, swertianolin, 8-*C*-glucopyranosyleriodictylol, longispinogenin), quercetin and their binding site on the domain A of *α*-glucosidase enzyme (3A4A).

**Figure 4 metabolites-12-01267-f004:**
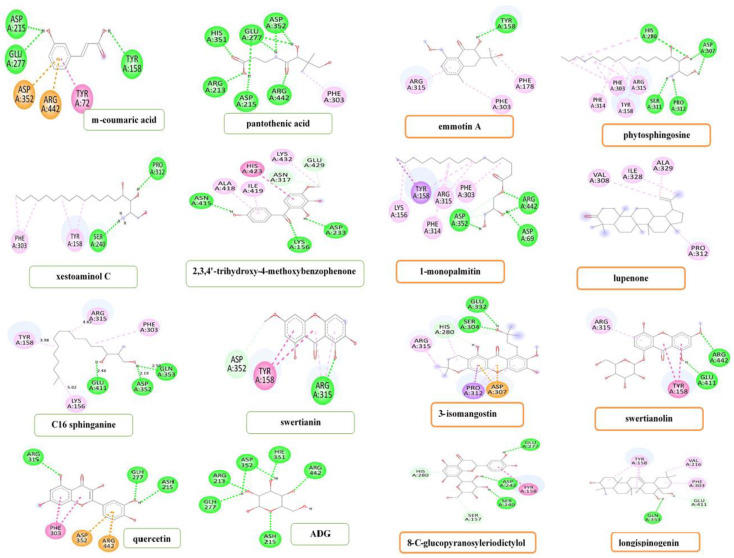
The 2D diagram of the α-glucosidase (3A4A) and 14 compounds, quercetin and ADG binding interactions.

**Figure 5 metabolites-12-01267-f005:**
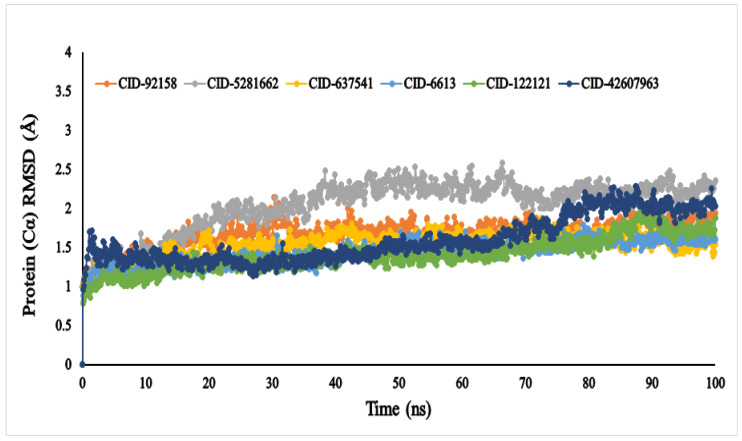
The RMSD values of the selected six compounds, viz., lupenone (CID: 92158), swertianolin (CID: 5281662), *m*-coumaric acid (CID: 637541), pantothenic acid (CID: 6613), phytosphingosine (CID: 122121), and 8-*C*-glucopyranosyleriodictylol (CID: 42607963), in association with the targeted protein *α*-glucosidase (PDB ID: 3A4A) are depicted by specific colour.

**Figure 6 metabolites-12-01267-f006:**
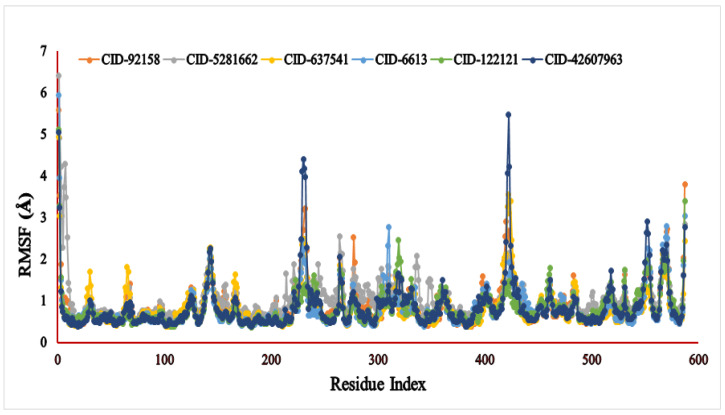
The RMSF values of the six selected compounds, viz., lupenone (CID: 92158), swertianolin (CID: 5281662), *m*-coumaric acid (CID: 637541), pantothenic acid (CID: 6613), phytosphingosine (CID: 122121), and 8-*C*-glucopyranosyleriodictylol (CID: 42607963), in association with the targeted protein *α*-glucosidase (PDB ID: 3A4A) are shown by specific colour.

**Figure 7 metabolites-12-01267-f007:**
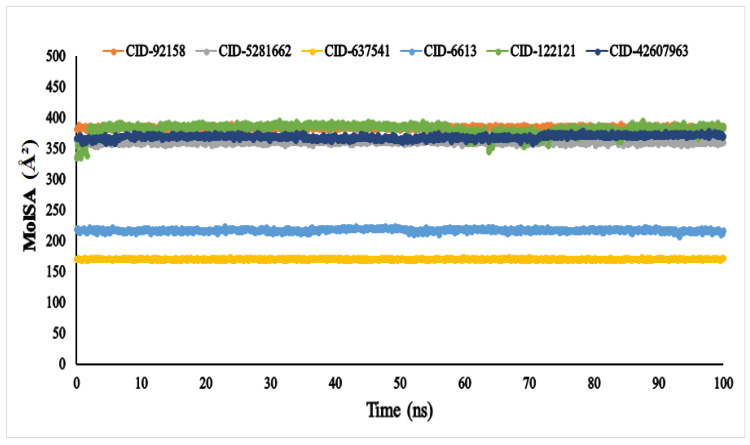
The molecular surface area (MolSA) of the of the selected six compounds, viz., lupenone (CID: 92158), swertianolin (CID: 5281662), *m*-coumaric acid (CID: 637541), pantothenic acid (CID: 6613), phytosphingosine (CID: 122121), and 8-*C*-glucopyranosyleriodictylol (CID: 42607963), in association with the targeted protein *α*-glucosidase (PDB ID: 3A4A) are shown by specific colour.

**Figure 8 metabolites-12-01267-f008:**
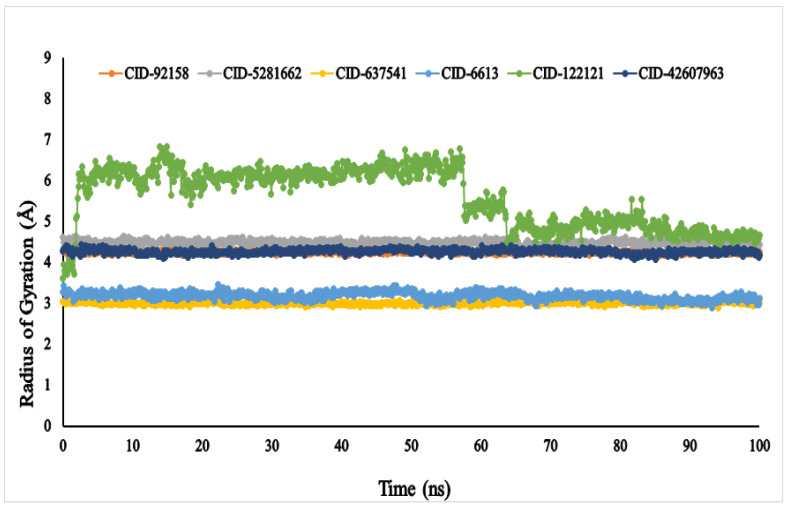
The radius of gyration (Rg) of the selected six compounds, viz., lupenone (CID: 92158), swertianolin (CID: 5281662), *m*-coumaric acid (CID: 637541), pantothenic acid (CID: 6613), phytosphingosine (CID: 122121), and 8-*C*-glucopyranosyleriodictylol (CID: 42607963), in association with the targeted protein *α*-glucosidase (PDB ID: 3A4A) are shown by specific colour.

**Figure 9 metabolites-12-01267-f009:**
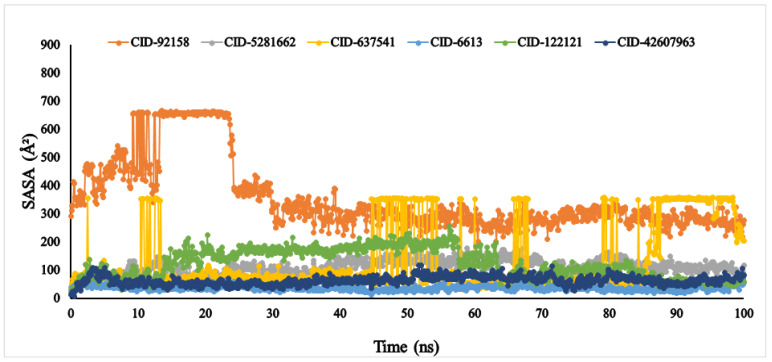
The solvent accessible surface area (SASA) of the selected six compounds, viz., lupenone (CID: 92158), swertianolin (CID: 5281662), *m*-coumaric acid (CID: 637541), pantothenic acid (CID: 6613), phytosphingosine (CID: 122121), and 8-*C*-glucopyranosyleriodictylol (CID: 42607963), in association with the targeted protein *α*-glucosidase (PDB ID: 3A4A) are shown by specific colour.

**Figure 10 metabolites-12-01267-f010:**
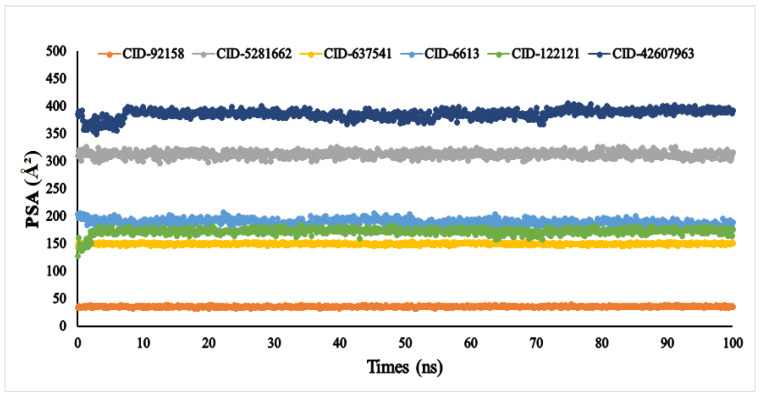
The polar surface area (PSA) of the selected six compounds, viz., lupenone (CID: 92158), swertianolin (CID: 5281662), *m*-coumaric acid (CID: 637541), pantothenic acid (CID: 6613), phytosphingosine (CID: 122121), and 8-*C*-glucopyranosyleriodictylol (CID: 42607963), in association with the targeted protein *α*-glucosidase (PDB ID: 3A4A) are shown by specific colour.

**Figure 11 metabolites-12-01267-f011:**
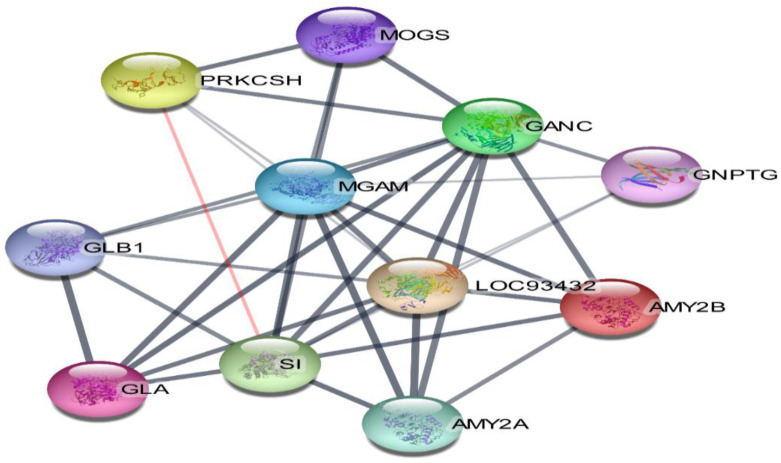
Network of protein interaction.

**Figure 12 metabolites-12-01267-f012:**
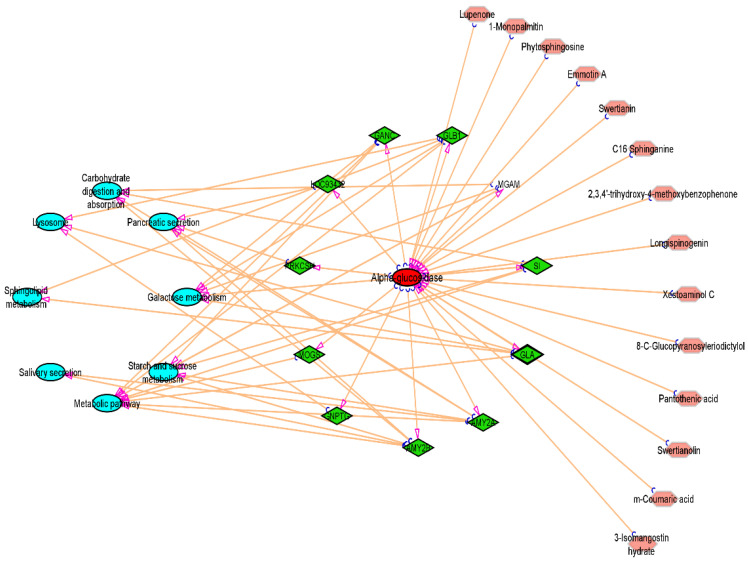
Compound-Target-Pathway Network.

**Figure 13 metabolites-12-01267-f013:**
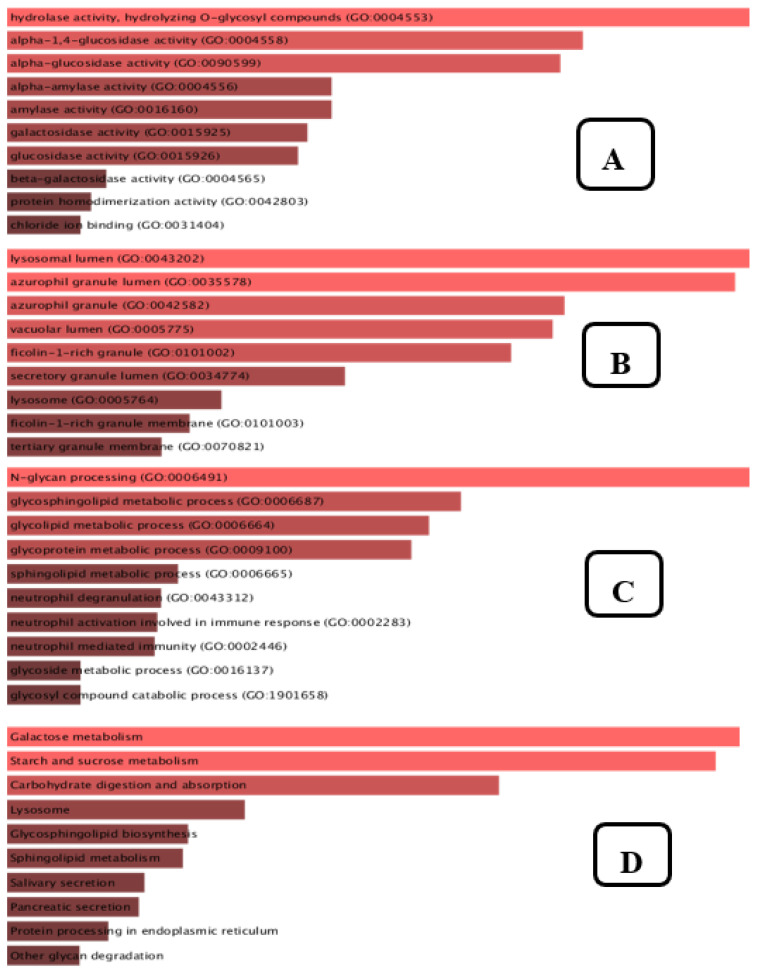
GO Molecular function (**A**), GO cellular component (**B**), GO Biological Process (**C**), KEGG pathway 2021 (**D**).

**Table 1 metabolites-12-01267-t001:** DPPH scavenging activity of the experimental extracts.

Concentration µg/mL	SCE-1 (%) Mean ± SEM	SCE-2 (%) Mean ± SEM	HRE (%) Mean ± SEM	Ascorbic Acid (%) Mean ± SEM
400	61.49 ± 0.25	75.36 ± 0.82	63.63 ± 0.64	98.00 ± 0.46
200	51.25 ± 0.57	61.34 ± 0.45	53.84 ± 1.73	92.83 ± 0.85
100	42.28 ± 1.39	44.43 ± 1.04	43.26 ± 0.95	87.49 ± 0.43
50	37.84 ± 0.19	36.56 ± 0.71	39.16 ± 0.95	70.99 ± 1.18
25	29.60 ± 0.70	26.58 ± 0.87	31.53 ± 1.61	54.06 ± 1.16
12.5	27.87 ± 2.00	23.75 ± 0.39	26.40 ± 2.21	46.21 ± 0.66

**Table 2 metabolites-12-01267-t002:** *α*-glucosidase inhibition activity of extracts obtained from different extraction techniques.

Concentration (μg/mL)	SCE-1 (%) Mean ± SEM	SCE-2 (%) Mean ± SEM	HRE (%) Mean ± SEM	Quercetin (%) Mean ± SEM
80.0	73.21 ± 0.31	81.79 ± 0.82	67.57 ± 0.68	91.53 ± 0.24
40.0	65.47 ± 0.62	68.16 ± 0.16	59.24 ± 0.49	79.48 ± 0.52
20.0	54.70 ± 0.51	60.97 ± 0.49	52.08 ± 0.21	64.43 ± 0.15
10.0	43.46 ± 0.71	53.47 ± 0.15	40.35 ± 0.28	61.09 ± 0.28
5.0	31.16 ± 0.41	48.39 ± 0.18	26.33 ± 0.20	54.21 ± 0.30
2.5	20.22 ± 0.98	37.28 ± 0.92	22.56 ± 0.18	43.70 ± 0.36
1.25	8.911 ± 0.31	32.07 ± 0.45	8.68 ± 0.28	35.62 ± 0.43

**Table 3 metabolites-12-01267-t003:** Bioactive compounds identified by Q-ToF-LCMS.

Identified Compounds	Formula	M/Z	Mass	Retention Time	Score
*m*-Coumaric acid	C_9_H_8_O_3_	182.080	164.046	1.012	98.35
Pantothenic acid	C_9_H_17_NO_5_	220.118	219.110	2.33	99.41
Xestoaminol C	C_14_ H_31_NO	230.248	229.241	12.193	97.9
2,3,4′-Trihydroxy-4-Methoxybenzophenone	C_14_H_12_O_5_	261.076	260.069	10.447	98.66
C16 Sphinganine	C_16_H_35_NO_2_	274.274	273.267	12.131	98.49
Swertianin	C_14_ H_10_ O_6_	275.054	274.047	9.079	99.39
Emmotin A	C_16_ H_22_ O_4_	279.160	278.152	16.76	95.7
Phytosphingosine	C_18_H_39_NO_3_	318.300	317.292	12.219	99.64
1-Monopalmitin	C_19_ H_38_O_4_	331.283	330.276	19.288	95.23
Lupenone	C_30_H_48_O	425.377	424.370	22.004	99.61
3-Isomangostin hydrate	C_24_H_28_O_7_	429.191	428.183	12.998	99.35
Swertianolin	C_20_H_20_O_11_	437.108	436.100	9.079	99.68
8-*C*-Glucopyranosyleriodictylol	C_21_H_22_O_11_	451.123	450.116	10.572	99.32
Longispinogenin	C_30_H_50_O_3_	476.409	458.376	22.185	99.2

**Table 4 metabolites-12-01267-t004:** Physiochemical and pharmacokinetics profile of all the identified compounds.

Physiochemical Properties										Pharmacokinetic Criteria					
Com.	MW	LogP	HA	HD	NRB	MR	SA	NL	DL	IA	BBA	AT	LD_50_	Class	HT
**1**	164.16	1.49	2	2	2	45.13	69.59	0	Yes	92.86	Yes	No	2980	5	No
**2**	219.24	−1.04	4	4	6	52.21	87.91	0	Yes	30.44	No	No	10000	6	No
**3**	229.41	3.62	2	2	11	73.28	101.62	0	Yes	90.37	Yes	No	3500	5	No
**4**	260.24	2.04	5	3	3	68.88	108.88	0	Yes	93.67	No	No	2000	4	No
**5**	273.46	3.37	3	3	14	84.06	119.14	0	Yes	91.79	Yes	No	3500	5	No
**6**	274.23	2.07	6	3	1	72.55	111.62	0	Yes	77.18	No	Yes	4000	5	No
**7**	278.34	1.63	4	2	3	76.90	118.94	0	Yes	95.02	Yes	Yes	2500	5	No
**8**	317.51	3.12	4	4	16	94.83	136.67	0	Yes	94.24	No	No	3500	5	No
**9**	330.50	4.36	4	2	17	97.06	142.17	0	Yes	90.92	Yes	No	5000	5	No
**10**	424.71	4.54	1	0	1	129.18	191.77	0	Yes	98.47	No	No	5000	5	No
**11**	428.48	4.17	7	3	4	119.72	179.38	0	Yes	92.34	No	Yes	550	4	No
**12**	436.37	−0.46	11	6	4	104.67	173.42	2	No	49.54	No	No	5000	5	No
**13**	450.40	−0.27	11	8	3	106.21	180.50	2	No	33.39	No	No	2000	4	No
**14**	458.73	6.11	3	3	1	137.21	201.99	1	Yes	90.18	No	No	4300	5	No

Legends: (**1**) *m*-coumaric acid; (**2**) pantothenic acid; (**3**) xestoaminol C; (**4**) 2,3,4′-trihydroxy-4-methoxybenzophenone; (**5**) C16 sphinganine; (**6**) swertianin; (**7**) emmotin A; (**8**) phytosphingosine; (**9**) 1-monopalmitin; (**10**) lupenone; (**11**) 3-isomangostin hydrate; (**12**) swertianolin; (**13**) 8-*C*-glucopyranosyleriodictylol; (**14**) longispinogenin; MW (molecular weight, g/mol); HA (hydrogen bond acceptor); HD (hydrogen bond donor); LogP (Predicted octanol/water partition coefficient); NRB (No. of rotatable bonds); IA (Intestinal absorption, % absorbed); LD_50_ (Median lethal dose); BBB (Blood Brain Barrier); HT (Hepatotoxicity); AT (AMES toxicity); NL (No. of Lipinski’s rule violations); DL (Drug-likeness).

**Table 5 metabolites-12-01267-t005:** Binding affinity values of *α*-glucosidase enzyme with the identified compounds determined by Q-ToF-LCMS.

Compounds	Binding Affinity (kcal/mol)
Control ligand (ADG)	−6.0
Quercetin	−8.4
*m*-Coumaric acid	−7.0
Pantothenic acid	−6.6
Xestoaminol C	−5.5
2,3,4′-Trihydroxy-4-Methoxybenzophenone	−7.8
C16 Sphinganine	−6.0
Swertianin	−7.9
Emmotin A	−7.2
Phytosphingosine	−5.8
1-Monopalmitin	−6.2
Lupenone	−9.5
3-Isomangostin hydrate	−9.0
Swertianolin	−9.2
8-*C*-Glucopyranosyleriodictylol	−9.6
Longispinogenin	−8.0

**Table 6 metabolites-12-01267-t006:** Identified Q-ToF-LCMS compounds and their amino acid residues, bond type and bond distance.

Compounds	Interacting Amino Acid Residues	Bond Type	Bond Distance (Å)
*m*-Coumaric acid	TYR158	Hydrogen bond	2.54
	ASP215	Hydrogen bond	1.96
	GLU277	Hydrogen bond	2.66
	ARG442	Pi-cation	3.69
	ASP352	Pi-anion	4.54
	TYR72	Pi-Pi-T shaped	5.47
Pantothenic acid	ARG442	Hydrogen bond	2.66
	ASP215	Hydrogen bond	1.99
	ARG213	Hydrogen bond	2.18
	HIS351	Hydrogen bond	1.94
	GLU277	Hydrogen bond	2.66, 2.66, 2.19
	ASP352	Hydrogen bond	2.16, 2.99
	PHE303	Pi-alkyl interaction	4.87
Xestoaminol C	PRO312	Hydrogen bond	2.70
	SER240	Hydrogen bond	2.45
	TYR158	Pi-alkyl interaction	5.0, 4.97
	PHE303	Pi-alkyl interaction	4.84, 5.10
2,3,4′-Trihydroxy-4-Methoxybenzophenone	ASP233	Hydrogen bond	2.68
	LYS156	Hydrogen bond	2.85
	ASN415	Hydrogen bond	2.53
	GLU429	Carbon-hydrogen bond	3.62
	ASN317	Carbon-hydrogen bond	3.06, 3.58
	HIS423	Pi-Pi-T-shaped interaction	5.16
	LYS432	Alkyl interaction	3.99
	ALA418	Pi-alkyl interaction	4.87
	ILE419	Pi-alkyl interaction	4.19
C16 Sphinganine	GLN353	Hydrogen bond	2.50
	ASP352	Hydrogen bond	2.19
	GLU411	Hydrogen bond	2.46
	LYS156	Alkyl interaction	5.02
	ARG315	Alkyl interaction	4.62
	PHE303	Pi-alkyl interaction	5.43
	TYR158	Pi-alkyl interaction	3.98
Swertianin	ASP352	Carbon-hydrogen bond	3.59
	ARG315	Hydrogen bond	2.05
	ARG315	Pi-alkyl interaction	4.52, 4.89
	TYR158	Pi-Pi-T-shaped interaction	4.95, 5.52
Emmotin A	TYR158	Hydrogen bond	2.40
	PHE178	Pi-alkyl interaction	5.16
	PHE303	Pi-alkyl interaction	4.45
	ARG315	Pi-alkyl interaction	4.38
Phytosphingosine	HIS280	Hydrogen bond	2.72
	ASP307	Hydrogen bond	2.19, 2.45
	PRO312	Hydrogen bond	2.20
	SER311	Hydrogen bond	2.46
	ARG315	Alkyl interaction	4.81, 4.84
	TYR158	Pi-alkyl interaction	4.78, 5.41
	PHE303	Pi-alkyl interaction	5.03
	PHE314	Pi-alkyl interaction	5.48
	HIS280	Pi-alkyl interaction	5.47
1-Monopalmitin	ARG442	Hydrogen bond	2.38, 5.98
	ASP69	Hydrogen bond	2.50
	ASP352	Hydrogen bond	2.52
	ASP352	Carbon-hydrogen bond	3.55, 3.61
	ASP303	Pi-alkyl interaction	5.08
	PHE314	Pi-alkyl interaction	5.46
	ARG315	Alkyl interaction	4.11, 5.06
	LYS156	Alkyl interaction	4.20
	TYR158	Pi-sigma interaction	3.60
	TYR158	Pi-alkyl interaction	4.58, 5.18
Lupenone	VAL308	Alkyl interaction	5.29
	ILE328	Alkyl interaction	5.29
	ALA329	Alkyl interaction	3.59
	PRO312	Alkyl interaction	4.55
3-Isomangostin hydrate	GLU332	Hydrogen bond	2.54
	SER304	Hydrogen bond	2.16
	HIS280	Pi-donor hydrogen bond	2.66
	HIS280	Pi-alkyl interaction	5.29
	ARG315	Alkyl interaction	4.33, 4.69
	PRO312	Pi-sigma interaction	3.69
	ASP307	Pi-anion interaction	3.79, 3.84
Swertianolin	ARG442	Hydrogen bond	2.92
	GLU411	Hydrogen bond	2.35
	TYR158	Pi-Pi-T-shaped interaction	5.50, 5.53
	ARG315	Pi-alkyl interaction	4.74
8-*C*-Glucopyranosyleriodictylol	GLU277	Hydrogen bond	2.78
	ASP242	Hydrogen bond	1.89
	SER240	Hydrogen bond	2.36
	SER157	Carbon-hydrogen bond	3.38
	HIS280	Pi-donor hydrogen bond	3.21
	TYR158	Pi-Pi-T-shaped interaction	4.99
Longispinogenin	GLN353	Hydrogen bond	2.91
	GLU411	Carbon hydrogen bond	3.55
	VAL216	Alkyl interaction	4.76
	PHE303	Pi-alkyl interaction	5.36
	TYR158	Pi-alkyl interaction	5.41, 4.59, 4.64
Quercetin	ARG315	Hydrogen bond	2.73
	GLH277	Hydrogen bond	2.01
	ASH215	Hydrogen bond	2.46
	PHE303	Pi-Pi T-shaped	4.96, 5.13
	ARG442	Pi-cation	3.75
	ASP352	Pi-anion	4.28
Control ligand (ADG)	ARG213	Hydrogen bond	1.80
	GLH277	Hydrogen bond	2.13
	ASH215	Hydrogen bond	2.44
	ASP352	Hydrogen bond	2.18, 2.37
	HIE351	Hydrogen bond	1.96
	ARG442	Hydrogen bond	1.90

## Data Availability

Data are available in the article.

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
