# Peer review of "In Vitro, In Silico and Network Pharmacology Mechanistic Approach to Investigate the α-Glucosidase Inhibitors Identified by Q-ToF-LCMS from Phaleria macrocarpa Fruit Subcritical CO2 Extract"

_metabolites, 2022, doi:10.3390/metabo12121267_

Round 1
Reviewer 1 Report
Dear authors
Article is perfect.
1. Title is long.
2. Scientific names as italic please.
3. Table format must be corrected (in lines).
Good luck

Author Response
Reviewer comments -1
Comment No. 1: Title is long.
Authors feedback: Done. The title length has been shortened as per the suggestion and the new title is shown below. “In vitro, in silico and network pharmacology mechanistic approach to investigate the α-glucosidase inhibitors identified by Q-ToF-LCMS from Phaleria macrocarpa fruits subcritical CO2 extract”.
Comment No. 2: Scientific names as italic please, reduce paragraph, remove underline.
Authors feedback: Corrected accordingly and highlighted.
Comment No. 3: Table format must be corrected (in lines).
Authors feedback: Table format has been corrected accordingly.

Reviewer 2 Report
Reviewer comments and suggestions
Manuscript ID: metabolites-2057738
In this paper, the authors studied the potential of different extracts obtained from Phaleria macrocarpa fruits having α-glucosidase inhibitors. The extracts were prepared through conventional and non-conventional extraction techniques. When this enzyme is inhibited, the rate of carbohydrate digestion slows down, resulting in less glucose absorption when the carbohydrates are not broken being very important for diabetic patients.
The use of traditional medicinal plant-based products may be an excellent option for treating or controlling diabetes without any adverse side effects, because of its characteristics like antioxidant and α-glucosidase inhibitor effects.
These studies and all the discussion around this theme are important because it has multiple potential health benefits.
In my opinion, the manuscript is suitable for publication in Metabolites after revision, because some information should be completed and improved.
General comments on the whole text:
- Check the text to the end of the line do not leave alone digits or values separated from the unit.
- Check the text in terms of language, to constantly keep one time.
- The English language needs correction and improvement. Some words are not correctly written, such as “ascorbic acid”.
Some concrete comments would be as follows:
- The abstract must clearly state the originality of this paper and the impact of this work on this field. Please, insert some relevant numeric results.
- The importance of this paper must be strongly supported as compared with previous publications, showing the added value of this publication.
-The experimental design has to be better explained. Include some agronomic information about the origin of fruits and some biometric information about fruits (mean weight); degree of ripening, ….. From how many plants were collected fruits? From a total of how many fruits per plant you have obtained the extracts.
- The authors must emphasize the importance of the use of subCO2 extraction. Which advantages it has when compared with other techniques?
- What is the relation between antioxidant activity and α-glucosidase inhibition?
- The authors have to explain the following sentence (page 22) “That’s why we considered all the hit compounds for in silico molecular docking to know their mechanism of action”.
- The authors must improve the discussion and the validation of conclusions. The limitations of this study must be presented.
- What are the quantification and detection limits of the analytical methodology?
- The Materials and Methods section has to be completed because the authors do not describe the conditions for heating under reflux extract (HRE).
- Was any experiment design used?
- The authors have to explore the importance of in vivo studies.
Final comments and considerations: It deserves to be published at Metabolites after the suggestions and corrections listed above are amended. Authors should be aware that their study has several limitations because improving an experiment design is very important and essential for final conclusions.
Author Response
Dear Respected 23-11-2022
Guest Editor
Dr. Prawej Ansari
Special issue “Frontiers of Natural Antidiabetic Drug Discovery"; Metabolites
Subject: Revised research manuscript (2057738) entitled “In silico molecular dynamic simulations and network pharmacology mechanistic approach to investigate the α-glucosidase potential of Phaleria macrocarpa fruits subcritical carbon dioxide extract compounds identified by Q-ToF-LCMS analysis”.
Thank you very much for giving us an opportunity to improve our research manuscript based on the constructive comments made by all reviewers in favour of our research work. We have incorporated all sorts of necessary corrections to our utmost best in regard to improve our article accordingly. Hence, we hope that now our research manuscript is complete and suitable in its current form for the consideration of possible publication in your esteemed journal. Please find attached the authors response to all the comments.
We have also shortened the title of our research manuscript based on the suggestion made by the reviewer 1 and it is given below for your kind perusal as well:
New Title: “In vitro, in silico and network pharmacology mechanistic approach to investigate the α-glucosidase inhibitors identified by Q-ToF-LCMS from Phaleria macrocarpa fruits subcritical CO2 extract”.
Looking forward to hearing from you soon. Thanks
With my highest regards,
Sincerely yours
Reviewer comments -2
Comment No. 1: Check the text to the end of the line do not leave alone digits or values separated from the unit
Author's feedback: Done and corrected throughout the research manuscript.
Comment No. 2: Check the text in terms of language, to constantly keep one time.
Author's feedback: Checked and corrected throughout.
Comment No. 3: The English language needs correction and improvement. Some words are not correctly written, such as “ascorbic acid”.
Author's feedback: All language errors have been carefully corrected and highlighted.
Comment No. 4: The abstract must clearly state the originality of this paper and the impact of this work on this field. Please, insert some relevant numeric results.
Author's feedback: Done accordingly. Thanks for the constructive comments to improve our article further. We have improved our abstract as per the comments and suggestions made by you. All additions have been highlighted for your kind perusal.
Comment No. 5: The importance of this paper must be strongly supported as compared with previous publications, showing the added value of this publication.
Author's feedback: Thanks for the valuable comment. We really appreciate it. Hence, we have included the previous publications data in the discussion section on page 22, to support our findings as well as to increase the value of our findings and highlighted.
Comment No. 6: The experimental design has to be better explained. Include some agronomic information about the origin of fruits and some biometric information about fruits (mean weight); degree of ripening, ….. From how many plants were collected fruits? From a total of how many fruits per plant you have obtained the extracts.
Author's feedback: Done. Added and highlighted on page 25.
Comment No. 7: The authors must emphasize the importance of the use of subCO2 extraction. Which advantages it has when compared with other techniques?
Author's feedback: Added and highlighted on page 21.
Comment No. 8: What is the relation between antioxidant activity and α-glucosidase inhibition? The authors have to explain the following sentence (page 22) “That’s why we considered all the hit compounds for in silico molecular docking to know their mechanism of action”.
Author's feedback: Explained and highlighted on page 22.
Comment No. 9: The authors must improve the discussion and the validation of conclusions. The limitations of this study must be presented.
Author's feedback: Updated and highlighted on page 29. The limitations of this study have been mentioned and highlighted.
Comment No. 10: What are the quantification and detection limits of the analytical methodology?
Author's feedback: We have mentioned in the methodology and followed the lab-established method in our study.
Comment No. 11: The Materials and Methods section has to be completed because the authors do not describe the conditions for heating under reflux extract (HRE).
Author's feedback: Done. We have described and highlighted on page 27.
Comment No. 12: Was any experiment design used?
Author's feedback: Yes, added at line 705 on page 27.
Comment No. 13: The authors have to explore the importance of in vivo studies.
Author's feedback: Done and added. We have added our future research work to further confirm the antidiabetic potential of this plant through in vivo study.

Round 2
Reviewer 2 Report
The authors replied to my comments and they have provided a new and improved version of the paper.
Final comments and considerations: In my opinion, the manuscript is suitable for publication in Metabolites.
Author Response
We have incorporated all corrections as per your advice and also revised the structure order of our manuscript and rechecked the reference citation order throughout.